# Identifying Patients with Familial Chylomicronemia Syndrome Using FCS Score-Based Data Mining Methods

**DOI:** 10.3390/jcm11154311

**Published:** 2022-07-25

**Authors:** Ákos Németh, Mariann Harangi, Bálint Daróczy, Lilla Juhász, György Paragh, Péter Fülöp

**Affiliations:** 1Division of Metabolic Disorders, Department of Internal Medicine, Faculty of Medicine, University of Debrecen, H-4032 Debrecen, Hungary; akos.nemeth@gmail.com (Á.N.); harangi@belklinika.com (M.H.); seber-juhasz.lilla@med.unideb.hu (L.J.); paragh@belklinika.com (G.P.); 2Doctoral School of Health Sciences, University of Debrecen, H-4032 Debrecen, Hungary; 3Aesculab Medical Solutions, Black Horse Group Ltd., H-4029 Debrecen, Hungary; 4Institute for Computer Science and Control (SZTAKI), Eötvös Loránd Research Network, H-1111 Budapest, Hungary; daroczyb@gmail.com; 5Department of Mathematical Engineering (INMA/ICTEAM), Université Catholique de Louvain, 1348 Louvain-la-Neuve, Belgium

**Keywords:** data mining, familial chylomicronemia syndrome, FCS score, machine learning, screening

## Abstract

Background: There are no exact data about the prevalence of familial chylomicronemia syndrome (FCS) in Central Europe. We aimed to identify FCS patients using either the FCS score proposed by Moulin et al. or with data mining, and assessed the diagnostic applicability of the FCS score. Methods: Analyzing medical records of 1,342,124 patients, the FCS score of each patient was calculated. Based on the data of previously diagnosed FCS patients, we trained machine learning models to identify other features that may improve FCS score calculation. Results: We identified 26 patients with an FCS score of ≥10. From the trained models, boosting tree models and support vector machines performed the best for patient recognition with overall AUC above 0.95, while artificial neural networks accomplished above 0.8, indicating less efficacy. We identified laboratory features that can be considered as additions to the FCS score calculation. Conclusions: The estimated prevalence of FCS was 19.4 per million in our region, which exceeds the prevalence data of other European countries. Analysis of larger regional and country-wide data might increase the number of FCS cases. Although FCS score is an excellent tool in identifying potential FCS patients, consideration of some other features may improve its accuracy.

## 1. Introduction

Fasting chylomicronemia may rarely be due to a monogenic disorder that markedly reduces the activity of lipoprotein lipase (LPL), resulting in a decreased clearance of the triglyceride-rich lipoproteins from plasma [1]. This condition, referred to as familial chylomicronemia syndrome (FCS), is characterized by severe hypertriglyceridemia and sustained fasting chylomicronemia, thus predisposing affected individuals to recurrent episodes of pancreatitis. With an estimated frequency of one per million in the population, FCS is usually due to the homozygous or compound heterozygous mutations of the *LPL* gene, leading to a severe lack of functioning LPL protein [2]. Although, the majority of the FCS patients are carriers of loss-of-function mutations in the *LPL* gene, similar mutations are found to be causal in FCS, including apolipoproteins C2 and A5 (*APOC2* and *APOA5*, respectively), lipase maturation factor 1 (*LMF1*), glycosylphosphatidylinositol-anchored high-density lipoprotein-binding protein 1 (*GPIHBP1*) and glycerol-3-phosphate dehydrogenase 1 (*G3PDH1*) [3,4,5,6]. 

Compared to those with multifactorial chylomicronemia syndrome (MFCS), patients with FCS are usually younger and less likely to possess any of the aggravating factors of hypertriglyceridemia; however, they are more prone to develop pancreatitis on the basis of the sustained chylomicronemia [7]. Interestingly, FCS patients are less likely to have cardiovascular disease (CVD), probably because of the severe reduction in LPL activity reducing the formation and accumulation of the atherogenic chylomicron and very low density lipoprotein (VLDL) remnants [2]. With a mortality rate of 2–5%, acute pancreatitis is the most dangerous consequence of hypertriglyceridemia [8]. Recently, an international expert panel proposed an excellent and easy-to-use diagnostic tool named the FCS score (Table 1) for the better identification of FCS patients [6]. According to Moulin et al., the FCS score turned out to have a sensitivity of 88% and specificity of 85% in identifying individuals with “very likely FCS”. 

Although the disease represents a great health burden, exact data are lacking about the frequency of the disease in Hungary and other European countries as well [6]. Therefore, we aimed to identify FCS patients using the above mentioned FCS score with data mining methods in two major hospitals of the Northern Great Plain region of Hungary. We also tried to assess the usability of the FCS score using various machine learning methods that were trained on the data of previously identified FCS patients, individuals likely to have FCS based on their FCS score and the total clinical population in Debrecen (*n* = 590,500).

## 2. Materials and Methods

### 2.1. Patients and Methods

We obtained raw data from the hospital record system of the two leading medical centers of the Northern Great Plain region of Hungary including University of Debrecen Clinical Center (UDCC) and the County Hospital of Szabolcs-Szatmár-Bereg (CHSSB). Summing up eight total years, the data source contained all medical records from these two centers between 1 January, 2007 and 31 December, 2014. Through the servers of Aesculab Medical Solutions (Black Horse Group Ltd., Debrecen, Hungary), we accessed, cleaned, preprocessed and structured anonymous data that contained all medical records from these healthcare providers. As discussed previously [9], the studied population was considered to be representative for the regional population, therefore, the calculated prevalence may precisely estimate the regional prevalence of FCS. The information processed for the study contained three data sources as (i) laboratory data, (ii) diagnostic data using, and transforming to, the International Statistical Classification of Diseases and Related Health Problems (ICD)-10 convention and (iii) textual data including all hospital appointments. Data cleaning, preprocessing steps, detailed methodologies and software used were described previously [9]. The feature set (feature space) for the training included (i) all available nominal laboratory data during the medical history with nominal values calculated for the same units (e.g., triglycerides above 1.7 mmol/L) and (ii) the medical history either available from the diagnosis or mined from the textual data and calculated to 5 characters of the ICD-10, (e.g., E7800). The FCS score calculations and chart generation were performed with open-source software solutions on the textual data (Appendix A).

From the mined data, we calculated the previously proposed FCS score for each patient and grouped them according to the likelihood of FCS. Following data selection and screening, the medically evaluated data were trained with multiple machine learning techniques, including rectified linear unit neural networks (ReLU), adaptive boosting (AdaBoost), gradient boosting (XGBoost) and support vector machines (SVM). The training was carried out with an open source software (Appendix A) using the UDCC site clinical data. Tests were performed both on the trained data (with a 50–50 split) and on the CHSSB data as well. Labelling previously identified FCS patients as “positive” and individuals with no previous diagnosis of FCS as “negative”, we trained binary classification models on a dataset, which contained all previously identified FCS patients labeled as “positive”, and randomly selected patients from the remaining part of the clinical population labelled as “negative”. We also experimented with models trained on a dataset where we treated individuals likely to have FCS based on their FCS score as patients belonging to the “positive” label.

### 2.2. About Machine Learning

We may define the problem as a traditional binary classification as we have a finite, real valued descriptor and a binary label for each patient. Thus, a patient may either have FCS, thus labelled as “1”, or lack FCS and labeled as “0”. Based on the annotated dataset, several ways exist to identify relations between the features (including the elements of the descriptors that contain the ICD-10 diagnosis, as well as laboratory test values) and the known labels. In order to determine the best method for FCS classification and to approximate the performance of the models over the whole population, we built models using subsets of patients with known true labels as clinically diagnosed FCS, and evaluated the performance of the learned models on an independent dataset with known true labels. Our reasoning was based on the fundamental theory of generalization introduced by Vapnik and Chervonenkis in 1971 [10] and as a set of consequences of the theorem, which apply to all methods but a set of special neural networks. For the latter, we refer to Nagarajan and Kolter [11] and Devroye et al. [12]. Therefore, even if the bounds in the Vapnik–Chervonenkis generalization are not informative about deep neural networks on the first hand, there may be an underlying structure for which the theorem is meaningful in practice, too. There are three key rules based on the theorem, which are in shape with the fundamentals of data mining and machine learning: (i) prefer models with low complexity to provide capacity to learn any labeling [13], (ii) evaluate on an independent test set and (iii) use a training set as large as possible. 

To cover different but the most efficient methods, we selected three widely used machine learning frameworks, including tree ensembles (AdaBoost and XGBoost) [13,14], “shallow” neural networks with kernel functions (SVM) [15] and fully connected “deep” neural networks with ReLU activations [16]. In comparison to ReLU networks, tree ensembles methods are less powerful as a function approximation technique, while the smaller capacity helps in the case of small datasets like ours or non-spatiotemporal structural variables, when there are no previously known reoccurring structures over the features. The order of the features is arbitrary in our study as they do not form rigid structures, hence, we used the only viable option and adopted fully connected artificial neural networks. Tree ensembles and kernel-based methods are not sensitive to the order of the features.

Tree ensembles build a set of “weak” classifiers from small, almost random decision trees. There are several methods to determine the set of decision trees and their importance e.g., random forest [17], adaptive boosting [13] or gradient boosting machines [14]. In the case of the neural networks, we built fully connected deep networks with ANN (artificial neural network) that were trained using ReLU as activations, and the parameters were optimized with adaptive momentum [18]. Finally, SVM models were trained with various kernel functions, including linear, polynomial or radial basis functions. Table 2 indicates the best performing methods per class. 

Besides sensitivity, specificity and accuracy, the most important metric is area under the receiver operating characteristic curve (ROC AUC) as an evaluation method for our binary classification method. Sensitivity is measured as the proportion of true positives in patients with FCS, while specificity describes the proportion of true negatives in patients without FCS. Accuracy is the proportion of the total number of patients that are correctly identified in the studied population. ROC curve is defined by the point pairs of true positive rates (sensitivity) and false positive rates (1 minus specificity) at different threshold settings. AUC can be interpreted as the probability of classifying a positive sample with higher confidence than a negative sample.

It is important to note that, based on the trees learned by a gradient boosted tree model, it is possible to rank the features using their position in the trees. There are multiple methods ranging from the simple count of occurrence to a complex subset identification that may yield a generously good ranking of the features. We relied on a weighted version of the former, most commonly used method [19]. Additionally, the order of the trees learned during the boosting phase is of utmost importance, thus, we decided to investigate the learnings of the first couple of trees learned by the model.

## 3. Results

Based upon the features of the previously proposed FCS score, we calculated the score of each individual that visited the two major healthcare providers in our region during the study period (*n* = 1.341.722; mean age: 38.12 ± 23.37 years, male/female: 602.258/739.464; 45/55%). Patient characteristics and their calculated FCS score are listed on Table 3. We identified a total of 26 patients very likely with FCS (score ≥ 10). These data suggest that FCS might be more frequent, at least in our region, with an estimated prevalence of 19.4 per million.

For a rapid estimation of FCS scores, we gradually cut down data based on some strong key features of the score system to estimate the number of the patients that fell into the three major categories of “highly unlikely FCS”, “unlikely FCS” and “likely FCS”. As FCS is a disease characterized by serum triglyceride (TG) levels, we chose features which contributed markedly to the FCS score and were easily measurable with less subjectivity (Figure 1).

Therefore, we took patients with fasting TG levels exceeding 10 mmol/L for three consecutive cases (+5 points) and those who never had TG levels less than 2 mmol/L (thus avoiding the −5 points), and added those patients who had no secondary causes such as diabetes mellitus, metabolic syndrome, hypothyroidism, corticosteroid therapy or alcohol abuse (+2 points). To further enhance this estimation of FCS scores and find those that potentially live with undiagnosed FCS, we added key features of fasting TG levels exceeding 20 mmol/L at least once (+1 point), symptoms below 40 years (+1 point) and positive history of pancreatitis (+1 point). Key features in the two major healthcare providers (UDCC and CHSSB) for FCS score estimation and the number of the patients falling into the score categories are represented on Table 4, respectively. Some intra-regional difference was detectable as we estimated the prevalence of “likely FCS” to be 8.47 per million in UDCC and 5.32 in CHSSB, respectively.

As with the total population, we also calculated FCS score for every single patient available in the hospital database, separately in the two medical centers (Table 5, respectively). Based on our results, the calculated prevalence of FCS is 27.11 per million in the Debrecen (UDCC) region and 13.3 per million in the Nyíregyháza (CHSSB) region. Overall, male patients had a 4 to 5 times increased chance for a “likely FCS” than females. The magnitude of the number of patients with a calculated FCS score of 10+ (“likely FCS”) was comparable with the estimated prevalence when checking the patients individually.

As our estimated prevalence turned out to be one order of magnitude higher than the literature data, we decided to evaluate thoroughly those patients of UDCC with an estimated 7+ score (*n* = 275, see Table 3). Therefore, all patients of this medical center with an estimated score falling into “unlikely FCS” and “likely FCS” diagnoses underwent a detailed evaluation of their medical history, TG levels and clinical signs in order to find those with undiagnosed FCS. During this data revision, we identified 7 patients with FCS and, without genetic testing, marked an additional 14 individuals with potential FCS. These data indicate an estimated prevalence of 11.8–35.6 FCS patients per million, which is a similar magnitude to our calculation detailed above.

Then we utilized machine learning, which was trained and tested on the UDCC dataset to identify those FCS patients who had ever been hospitalized. As trained data, we used the above mentioned 7 confirmed and 14 potential FCS patients against those who scored 7+ in the FCS score system and against random individuals. The results of the mathematical modeling are depicted on Table 2, while model parameters are detailed in Appendix B. During classification, boosting models (i.e., AdaBoost and XGBoost) performed most successfully in terms of ROC/AUC measures, tightly followed by support vector machines. Deep neural networks lagged behind, notably in terms of overall performance. 

Table 6 shows the summarized importance of conditions of the history in defining FCS, using all model trainings. To evaluate the accuracy of the FCS score, we trained these confirmed and potential FCS patients vs. patients with 7+ FCS score. Individual laboratory measurements were mined from the medical histories of the patients with no absolute values assigned to them. The parameters were ranked by the mathematical models from 0 to 100, where the value of 100 indicates the most important condition in decision making. Our results confirmed the foundational importance of the TG levels, as (i) the highest TG level and (ii) the average TG level were found to be the most important features, while (iii) conditions characterizing deviations in the TG concentrations (i.e., TG fluctuation, as well as highest and lowest TG levels) were also among the top conditions of the history. Cholesterol level also turned out to be a substantial feature in defining FCS. These conditions are the most important ones to distinguish FCS patients from those with no FCS but high FCS score. 

To find the most important conditions and decisive laboratory cut values that can be used for population screening, we also trained machine learning using the data of the confirmed and potential FCS patients vs. all patients (Table 7). The cut values do not make distinction between their absolute importance but help the clinicians to get closer or away from the likelihood of FCS. Altogether, we found that patients may be identified based upon their highest and lowest TG levels, average TG levels and TG level deviations, as well as the highest and lowest total cholesterol concentrations and the deviations of the total cholesterol level. We also identified other parameters that may help to find individuals with potential FCS, as increasing hemoglobin, MCHC, basophil granulocyte, lymphocyte, or amylase above the cut levels raised the probability of FCS. On the other hand, elevated GPT, GGT, glucose, sodium and creatinine measurement cut levels decreased the chance of FCS. Interestingly, we also found that inflammatory markers as WBC and CRP, as well as the amylase activity had a negative impact on the probability of FCS.

## 4. Discussion

We suspected the regional frequency of FCS to be 19.4 per million among hospital goers, which exceeds the estimated worldwide prevalence of 1 per million [20]. As FCS is considered to be a rare disease, recent data indicate higher frequency of the disease when using larger cohorts. Indeed, reviewing the data of more than 1.5 million patients, Pallazola et al. found an FCS prevalence of 13 per million among the patients of a quaternary medical center [21]. On a smaller dataset of thirty thousand children, the prevalence of type 1 hyperlipoproteinemia (i.e., familial chylomicronemia syndrome) was estimated to be about 1 in 300,000 [22]. It is important to emphasize that we studied a population that was treated or checked in a hospital, which might have contributed to the variance of the disease prevalence. Though falling into the same magnitude, we also found the FCS prevalence to be different between the medical providers, either estimated with using key features of the disease or calculated individually in each patient. These discrepancies are presumably due to the different levels of care and the covered territories of the medical providers (university hospital vs. county hospital). Indeed, with its various lipid/metabolic disease outpatient clinics, our university hospital accepts patients from the county hospital, as well. More targeted history taking, wider diagnostic and laboratory availabilities may also explain our prevalence results after revisiting the university hospital data. Besides indicating the usability of our methods in distinct populations, our findings highlight the need of the specialist’s expertise in recognizing FCS.

The diagnosis of FCS is largely based upon genetic analysis and post-heparin LPL activity assay [7]. Recently, an expert panel of lipidologists proposed a very practical FCS scoring system for the better identification of patients with this rare, inherited disease [6]. A solid advantage of the FCS score is the strong reliance on the exact serum triglyceride measurements. Indeed, the selection of the potential patients can be reduced to 1–2‰, if studying those with TG levels exceeding 10 mmol/L for three consecutive occasions and never below 2 mmol/L (as indicated on Table 4). Adding the other strong and measurable condition (TG levels exceeding 20 mmol/L at least once) cut down the patient selection to the zone of ten thousandths (‱). 

On the other hand, we realized that patients with the highest FCS scores are not necessarily the similar ones that we diagnosed. That can be due to incomplete history taking (e.g., missing targeted questions on conditions aggravating hypertriglyceridemia), which can hamper proper diagnosis [23]; therefore, FCS scoring seems to be perfect when all such secondary factors can be excluded by the dedicated physician, while there could be an area for improvement when approaching FCS score on a larger, automatized level.

Machine learning, however, may serve as a helpful tool to better identify rare diseases when using larger datasets [9,24]. Trained and tested on the UDCC data, we also tried to find those FCS patients who, with any diagnosis, had ever been hospitalized in our university hospital. We found gradient boosting and SVM to be the most successful in terms of ROC/AUC measures. Contrary to neural networks, these boosting-based models were more useful to find those with FCS. Our investigations on laboratories indicated that mild-to-moderate or very high TG concentration cuts further improve identifying potential FCS patients, even when peaking above 20 mmol/L. Interestingly, total cholesterol level may also be a promising asset to improve identification. The role of cholesterol, however, seems to be more complex, as the likelihood of FCS decreases below 4 mmol/L and above 11 mmol/L. In other words, patients with low or with very high cholesterol levels should not be considered to have FCS, which indicates the importance the triglyceride-rich lipoprotein cholesterol and the intimate interplay between cholesterol and triglyceride metabolism [25]. 

On the other hand, we found several metabolic parameters including liver transaminases and serum glucose, whose increased activities or concentrations affected negatively the probability of FCS. These findings might be due to the common presence of insulin resistant conditions as obesity, type 2 diabetes mellitus and non-alcoholic fatty liver disease (NAFLD) among hospital goers and are concordant with the recent report of Paquette et al., who found higher activities of gamma-glutamyl transferase (GGT) in MFCS compared to FCS [7]. Of note, although occurring in both FCS and MFCS patients, NAFLD was observed to be significantly less frequent in patients with familial chylomicronemia syndrome [26].

Interestingly, we found that elevated amylase activity had a negative impact on FCS probability, which indicates a high prevalence of such laboratory findings in the studied population. Longitudinal studies on well-characterized patient populations, however, confirmed the higher incidence of acute pancreatitis in FCS patients [27]. These investigations may also shed light on cardiovascular outcomes in these subjects, as well. Nevertheless, besides indicating the potential existence of multifactorial backgrounds, our findings may also help to increase FCS awareness, as higher glucose levels or transaminase activities decrease the probability of FCS. 

Limitations also exist in our study. Hospital goers represent a population that differs from the normal population; therefore, our calculations might overestimate the frequency of the disease. Although we could study a relatively large cohort of patients, it did not directly represent the total population in our region, as not 100% of the population goes to hospital each year. Also, we were unable to assess the data about family history and did not perform genetic testing to diagnose FCS. Verifying the existence of confirmed or potentially pathogenic mutations in LPL or other genes modulating lipoprotein lipase activity would have contributed to improve identification of potential FCS patients in the studied population. Genetic analysis of gene variants with triglyceride-lowering effect would also have modified our results. In addition, hospital goers tended to be older and checked more frequently. On the contrary, younger patients usually had less thorough laboratory examinations and their history was less detailed and asked less frequently. Such tendencies bias the identification of FCS patients towards the elderly. Additionally, a larger population is needed to define those exact cuts in cholesterol levels that could improve FCS scoring. Although our machine learning models found their impact on the likelihood of FCS, the real-life importance of the other laboratory parameters should also be addressed in future studies. While machine learning may overestimate the incidence of FCS, it also may help to reduce the number of those individuals that would require expensive and time-consuming genetic analysis.

## 5. Conclusions

Using the previously proposed FCS scoring based on a large hospital database, we found an increased prevalence of familial chylomicronemia syndrome in our region. Data mining and machine learning seem to be promising tools in screening for FCS; however, further studies on larger, national or international datasets are of major importance to prove their accuracy and usefulness. Also, an analysis of larger populations might increase the number of discovered FCS cases.

Although FCS scoring is an easy-to-use tool to set FCS and MFCS apart, “fine tuning” of the features and inclusion of the total cholesterol levels may be considered to better identify FCS patients. Although the weight of cholesterol levels in the score has to be determined, this may alleviate the need for systematic genotyping in patients with severe hypertriglyceridemia and would also help identify the high-priority candidates for genetic analysis. Furthermore, early and accurate diagnosis is essential for effective treatment to avoid severe, life-threatening complications of FCS.

## Figures and Tables

**Figure 1 jcm-11-04311-f001:**
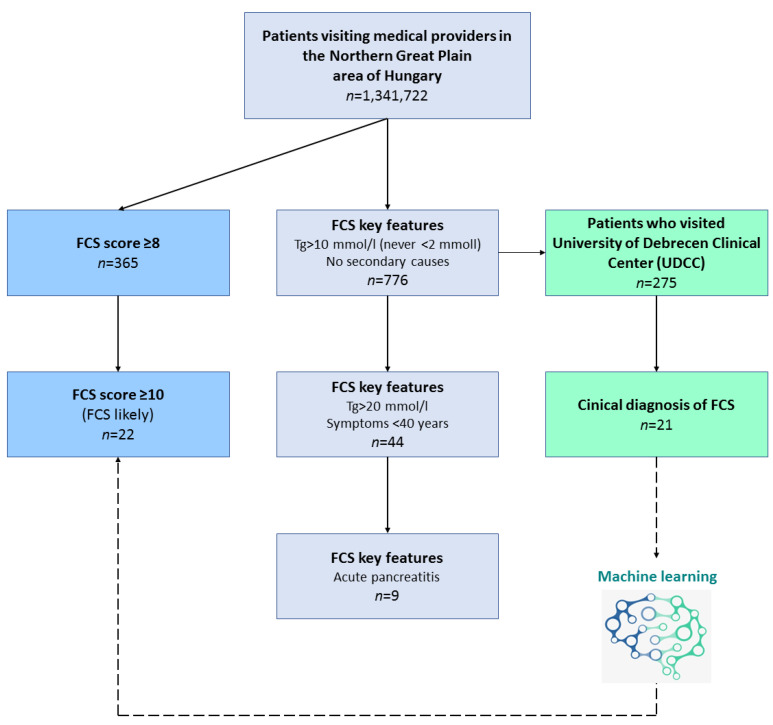
Flowchart of the rapid estimation of familial chylomicronemia (FCS) score.

**Table 1 jcm-11-04311-t001:** Familial chylomicronemia syndrome scoring, according to Moulin et al.

	**Score**
1. Fasting TGs > 10 mmol/L for three consecutive blood analyses	+5
Fasting TGs > 20 mmol/L at least once	+1
2. Previous TGs < 2 mmol/L	−5
3. No secondary factor (except pregnancy and ethinylestradiol)	+2
4. History of pancreatitis	+1
5. Unexplained recurrent abdominal pain	+1
6. No history of familial combined hyperlipidemia	+1
7. No response (TG decrease <20%) to hypolipidemic treatment	+1
8. Onset of symptoms at age: - <40 years - <20 years - <10 years	+1 +2 +3

Score > 10: FCS very likely; Score < 9: FCS unlikely; Score < 8: FCS very unlikely.

**Table 2 jcm-11-04311-t002:** Classification performance of models trained on FCS.

Training Set	Test Set	Method	Exp.	Mean AUC	Std AUC	Mean ACC	Std ACC	Mean Sens.	Std Sens.	Mean Spec.	Std Spec.
50% Exam.	Ind. 50% Exam.	ReLU	30	0.735	0.064	0.895	0.024	0.212	0.160	0.950	0.029
		SVM	30	0.792	0.054	0.927	0.013	0.0	0.0	0.999	0.001
		ADA	30	0.770	0.053	0.902	0.014	0.110	0.121	0.970	0.023
		XGB	30	0.810	0.042	0.909	0.018	0.070	0.104	0.976	0.025
50% Exam.	Ind. 50% Exam. UDCC 5000 patients w/o FCS	ReLU	30	0.599	0.088	0.857	0.112	0.237	0.184	0.859	0.113
		SVM	30	0.872	0.057	0.998	0.001	0.0	0.0	0.999	0.001
		ADA	30	0.824	0.092	0.996	0.002	0.110	0.121	0.999	0.002
		XGB	30	0.871	0.074	0.997	0.001	0.070	0.104	0.999	0.001
50% Exam. & UDCC 1000 patients w/o FCS	Ind. 50% Exam. UDCC 5000 patients w/o FCS	ReLU	30	0.906	0.041	0.997	0.001	0.245	0.142	0.999	0.011
		SVM	30	0.955	0.024	0.999	0.001	0.0	0.0	0.999	0.001
		ADA	30	0.923	0.051	0.996	0.002	0.110	0.121	0.999	0.001
		XGB	30	0.982	0.015	0.997	0.001	0.091	0.096	0.999	0.001

**Table 3 jcm-11-04311-t003:** Calculated familial chylomicronemia (FCS) scores of patients visiting medical providers in the Northern Great Plain area of Hungary (pcm = 1:100,000; ppm = 1:1,000,000).

Cluster	FCS Score	Male Patients	Female Patients	Total Patients	Percentage of Patients
Highly unlikely FCS	0+	602.258 (45%)	739.464 (55%)	1.341.722	100%
	1+	5.612 (56%)	4.334 (44%)	9.946	7.41‰
	2+	1.659 (75%)	558 (25%)	2.217	1.65‰
	3+	1.441 (75%)	493 (25%)	1.934	1.44‰
	4+	1.307 (74%)	461 (26%)	1.768	1.32‰
	5+	1.272 (74%)	453 (26%)	1.725	1.29‰
	6+	909 (78%)	254 (22%)	1.163	8.67‱
	7+	705 (79%)	182 (21%)	887	6.61‱
Unlikely FCS	8+	298 (82%)	67 (18%)	365	2.72‱
	9+	56 (81%)	13 (19%)	69	5.14 pcm
Likely FCS	10+	17 (77%)	5 (23%)	22	1.64 pcm
	11+	3 (75%)	1 (25%)	4	2.98 ppm

**Table 4 jcm-11-04311-t004:** Familial chylomicronemia (FCS) score estimation on key features.

**A. FCS score estimation on key features (UDCC, all patients *)**				
**Cluster**	**Feature**	**FCS Score**	**Number of Patients**	**Percentage of Patients**
Highly unlikely FCS	Clinical site patients	0+	590.500	100%
	TG 10+ mmol/L and TG never 2- mmol/L	5+	665	1.13‰
	No secondary medical factors **	7+	275	4.67‱
Unlikely FCS	TG 20+ mmol/L at least once	8+	85	1.44‱
	Symptoms below age 40	9+	24	4.06 pcm
Likely FCS	Treated with acute pancreatitis	10+	5	8.47 ppm
**B. FCS score estimation on key features (CHSSB, all patients *)**				
**Cluster**	**Key Condition**	**FCS Score**	**Number of Patients**	**Percentage of Patients**
Highly unlikely FCS	Clinical site patients	0+	751.624	100%
	TG 10+ mmol/L and TG never 2− mmol/L	5+	1.046	1.39 ‰
	No secondary medical factors **	7+	501	6.67‱
Unlikely FCS	TG 20+ mmol/L at least once	8+	93	1.23‱
	Symptoms below age 40	9+	20	2.66 pcm
Likely FCS	Treated with acute pancreatitis	10+	4	5.32 ppm

(A): * Patients who visited University of Debrecen Clinical Center (UDCC) at least once between 2007–2014; ** diabetes, metabolic syndrome, hypothyroidism, corticosteroid therapy, alcohol abuse. (B) * Patients who visited County Hospital of Szabolcs-Szatmár-Bereg (CHSSB) at least once between 2007–2014; ** diabetes, metabolic syndrome, hypothyroidism, corticosteroid therapy, alcohol abuse.

**Table 5 jcm-11-04311-t005:** Familial chylomicronemia (FCS) score calculation of individual patients.

Cluster	FCS Score	Males (*n*)	Females (*n*)	Total (*n*)	Percentage
**A. FCS score calculation of individual patients (UDCC, all patients *)**					
Highly unlikely FCS	0+	251.949 (43%)	338.149 (57%)	590.098	100%
	1+	2368 (53%)	2.108 (47%)	4.476	7.59‰
	2+	589 (74%)	208 (26%)	797	1.35‰
	3+	538 (73%)	198 (27%)	736	1.25‰
	4+	506 (73%)	188 (27%)	694	1.18‰
	5+	490 (73%)	183 (27%)	673	1.14‰
	6+	340 (76%)	107 (24%)	447	7.58‱
	7+	250 (78%)	71 (22%)	321	5.44‱
Unlikely FCS	8+	110 (77%)	32 (23%)	142	2.41‱
	9+	31 (82%)	7 (18%)	38	6.44 pcm
Likely FCS	10+	10 (77%)	3 (23%)	13	2.20 pcm
	11+	2 (67%)	1 (33%)	3	5.08 ppm
**B. FCS score calculation of individual patients (CHSSB, all patients *)**					
Highly unlikely FCS	0+	350.309 (47%)	401.315 (53%)	751.624	100%
	1+	3.244 (59%)	2.226 (41%)	5.470	7.28‰
	2+	1070 (75%)	350 (25%)	1.420	1.89‰
	3+	903 (75%)	295 (25%)	1.198	1.59‰
	4+	801 (75%)	273 (25%)	1.074	1.42‰
	5+	782 (74%)	270 (26%)	1.052	1.40‰
	6+	569 (79%)	147 (21%)	716	9.53‱
	7+	455 (80%)	111 (20%)	566	7.53‱
Unlikely FCS	8+	188 (84%)	35 (16%)	223	2.97‱
	9+	25 (81%)	6 (19%)	31	4.12 pcm
Likely FCS	10+	7 (78%)	2 (22%)	9	1.19 pcm
	11+	1(100%)	0 (0%)	1	1.33 ppm

(A) * Patients who visited University of Debrecen Clinical Center (UDCC) at least once between 2007–2014. (B) * Patients who visited County Hospital of Szabolcs-Szatmár-Bereg (CHSSB) at least once between 2007–2014.

**Table 6 jcm-11-04311-t006:** Importance of conditions of the history in defining FCS, using all model trainings (expressed in relative importance scores, in the fractions of the most important features).

Confirmed and Potential FCS Patients vs. Patients with FCS Score of 7+		Confirmed and Potential FCS Patients vs. Random Individuals	
**Condition**	**Importance**	**Condition**	**Importance**
Highest triglyceride	100	Average triglyceride	100
Average triglyceride	50	Highest triglyceride	70
Average cholesterol	25	Lowest triglyceride	40
Triglyceride fluctuation	20	Triglyceride fluctuation	35
Lowest triglyceride	17	Average cholesterol	30
Lowest carbamide	16	Highest cholesterol	25
Highest cholesterol	15	Lowest cholesterol	15
Average hemoglobin	14	Cholesterol fluctuation	15
Lowest glucose	12	Average hemoglobin	10
Average alkaline phosphatase	10	Glucose fluctuation	10

**Table 7 jcm-11-04311-t007:** Summary of the most decisive laboratory value cuts in machine learning models and their impact on getting closer to (+) or away (−) from likelihood of FCS.

Laboratory Parameter	Cut (>)	Impact
Triglyceride	30 mmol/L	+
Triglyceride	18 mmol/L	+
Triglyceride	6.5 mmol/L	+
Cholesterol	11 mmol/L	−
Cholesterol	6.5 mmol/L	+
Cholesterol	4.0 mmol/L	+
Hemoglobin	95 g/L	+
MCHC	330 (g/L)	+
Amylase	20 U/L	+
Basophile granulocyte	0.6%	+
Lymphocyte	20%	+
Sodium	145 mmol/L	−
White Blood Cell	6.5 G/L	−
Neutrophile granulocyte	65%	−
GPT	15 U/L	−
GPT	200 U/L	−
GGT	35 U/L	−
GGT	350 U/L	−
Creatinine	68 µmol/L	−
CRP	5.0 mg/L	−
Glucose (fasting)	6.0 mmol/L	−

## Data Availability

All data generated or analyzed during this study are included in this published article. All data generated or analyzed during the current study are available from the corresponding author on reasonable request.

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
