# Peer review of "Identifying Patients with Familial Chylomicronemia Syndrome Using FCS Score-Based Data Mining Methods"

_jcm, 2022, doi:10.3390/jcm11154311_

Round 1

Reviewer 1 Report

Dear authors, thank you for this manuscript. I have some thoughts and suggestions on your research. 

In your abstract, you refer to the prevalence of FCS in "Central Europe" and "Other European countries", but I could not find a reference in the introduction. Could you please provide one? 

You described FCS as "recently referred to". I allow myself to correct, that this is no "recent" fact. Also LOF-mutations in LPL are not casual but causal - this being just a minor correcting.

I recommend checking your wording on the "diagnosis of FCS" as the score is a screening tool indicating the probability of FCS. Diagnosis is made clinically or genetically. 

Did you verify the FCS genetically in the patients who were identified by the FCS score? This was not clear to me.

In Table 7 it is not clear to me, which additional information you want to communicate? The rating "+"/"-" gives no concrete information and I can't see the additional value of this table. Can you update and give more detailed information? How to interpret "+"/"-"?

In your discussion, you write "We found the regional frequency of FCS to be...". This is not true unless you confirmed the suspected diagnosis genetically. The score only allows you to suspect FCS in a patient with a sensitivity of 88% and a specificity of 85%.

Overall, thank you for the manuscript, as FCS is a very important subject that needs more research! Good luck!

Author Response

Dear Reviewer, thank you for your thorough review and helpful comments on our manuscript. We hereby answer your suggestions as follows:

  1. In your abstract, you refer to the prevalence of FCS in "Central Europe" and "Other European countries", but I could not find a reference in the introduction. Could you please provide one?

Answer: Thank you for the comment. Indeed, there are no exact data on FCS prevalence in these countries, so we hereby refer to the original article from Moulin et al. (ref. No. 6) as authors of this paper comprises many European countries including Western and Central-Eastern Europe.

Based upon your suggestion, we modified the sentence in the introduction as follows (lines 66-67):

“Although the disease represents a great health burden, exact data are lacking about the frequency of the disease in Hungary and other European countries, as well [6].”

  1. You described FCS as "recently referred to". I allow myself to correct, that this is no "recent" fact. Also LOF-mutations in LPL are not casual but causal - this being just a minor correcting.

Answer: You are absolutely right in both comments. We deleted the word “recent” from the sentence in line 37 and changed the wording to “causal” in line 44.

These sentences are now:

Lines 37-40: “This condition, referred to as familial chylomicronemia syndrome (FCS), is characterized by severe hypertriglyceridemia and sustained fasting chylomicronemia thus predisposing affected individuals to recurrent episodes of pancreatitis.”

Lines 42-48: “Although, the majority of the FCS patients are carriers of loss-of function mutations in the LPL gene, similar mutations are found to be causal in FCS including apolipoproteins C2 and A5 (APOC2 and APOA5, respectively), lipase maturation factor 1 (LMF1), glycosylphosphatidylinositol-anchored high-density lipoprotein-binding protein 1 (GPIHBP1) and glycerol-3-phosphate dehydrogenase 1 (G3PDH1) [3-6].”

  1. I recommend checking your wording on the "diagnosis of FCS" as the score is a screening tool indicating the probability of FCS. Diagnosis is made clinically or genetically. 

Answer: Thank you for this really helpful suggestion. As we completely agree with you, we changed the wording in those sentences where the “diagnosis of FCS” could have been misinterpreted.

These sentences are now as follows:

Lines 29-30: “Although FCS score is an excellent tool in identifying potential FCS patients, consideration of some other features may improve its accuracy.”

Lines 58-60: “According to Moulin et al., FCS score turned out to have a sensitivity of 88% and specificity of 85% in identifying individuals with “very likely FCS”.”

Lines 97-98: “From the mined data, we calculated the previously proposed FCS score for each patient and grouped them according to the likelihood of FCS.”

Lines 179-181: “We identified a total of 26 patients with a very likely FCS (score ≥10).”

Lines 223-224: “Overall, male patients had a 4 to 5 times increased chance for a “likely FCS” than females.”

Lines 277-279: “The cut values do not make distinction between their absolute importance but help the clinicians to get closer or away from the likelihood of FCS.”

Lines 282-285: “We also identified other parameters that may help to find individuals with potential FCS, as increasing hemoglobin, MCHC, basophil granulocyte, lymphocyte, or amylase above the cut levels raised the probability of FCS.”

Lines 290-291 (table 7, figure legend): “Table 7. Summary of the most decisive laboratory value cuts in machine learning models and their impact on getting closer to (+) or away from (-) the likelihood of FCS.”

Lines 335-340: “Our investigations on laboratories indicated that mild-to-moderate or very high TG concentration cuts further improve identifying potential FCS patients, even when peaking above 20 mmol/L. Interestingly, total cholesterol level may also be a promising asset to improve identification. The role of cholesterol, however, seems to be more complex, as the likelihood of FCS decreases below 4 mmol/L and above 11 mmol/L.”

Lines 347-349: “On the other hand, we found several metabolic parameters including liver transaminases and serum glucose, whose increased activities or concentrations affected negatively the probability of FCS.”

Lines 356-358: “Interestingly, we found that elevated amylase activity had a negative impact on the FCS probability, which indicates a high prevalence of such laboratory findings in the studied population.”

Lines 361-363: “Nevertheless, besides indicating the potential existence of multifactorial background, our findings may also help to increase FCS awareness as higher glucose levels or transaminase activities decrease the probability of FCS.”

Lines 380-383: “Although our machine learning models found their impact on the likelihood of FCS, real-life importance of the other laboratory parameters should also be addressed in future studies.”

  1. Did you verify the FCS genetically in the patients who were identified by the FCS score? This was not clear to me.

Answer: We are really thankful for your question. Indeed, we have started the genetic analysis of those 21 patients with the clinical diagnosis of FCS (depicted on figure 1); and screening for the mutations of the LPL, APOC2, APOA5, LMF1, GPIHBP1 and G3PDH1 genes are currently in progress. Completing the analysis of 9 individuals so far, we have found confirmed or potentially pathogenic mutations in 5 of them.

Based upon your valuable comment and the suggestion of the other reviewer, lack of genetic analysis was also added to the limitations of our study in the discussion, as follows (lines 369-374):

“Also, we were unable to assess the data about family history and did not perform genetic testing to diagnose FCS. Verifying the existence of confirmed or potentially pathogenic mutations in LPL or other genes modulating lipoprotein lipase activity would have contributed to improve identification of potential FCS patients in the studied population. Genetic analysis of gene variants with triglyceride lowering effect would also have modified our results.”

  1. In Table 7 it is not clear to me, which additional information you want to communicate? The rating "+"/"-" gives no concrete information and I can't see the additional value of this table. Can you update and give more detailed information? How to interpret “+”/"-"?

Thank you for this comment. These laboratory value cuts are characterized by absolute value numbers: e.g., triglyceride level above or below 30 mmol/L, therefore getting closer to (+) or away from (-) the likelihood of FCS. Using our mathematical models, these cuts were found to be most decisive to identify potential FCS patients. Although, the values of these cuts might have pathophysiological importance, further investigations are needed to explain these findings. It is also important to emphasize that these cuts are not in direct connection with the relative importance scores of Table 6.

  1. In your discussion, you write "We found the regional frequency of FCS to be...". This is not true unless you confirmed the suspected diagnosis genetically. The score only allows you to suspect FCS in a patient with a sensitivity of 88% and a specificity of 85%.

Answer: You are right, therefore we changed the wording in this sentence, as follows (lines 295-296):

“We suspected the regional frequency of FCS to be 19.4 per million among hospital goers, which exceeds the estimated worldwide prevalence of 1 per million [20].”

For your kind information, we also revised and restructured some sentences in our manuscript with a supervision of a native English speaker. Again, we are very thankful for your valuable and thorough review. We think your comments and suggestions really improved the manuscript.

Reviewer 2 Report

The authors report their attempt to screen a large patient population for LAL Def, a rare genetic disorder of TG metabolism. They utilized a previously published FCS scoring system. Their research design and methodology appears appropriate, and the article well written. A few grammatical changes would improve the flow and readability of their manuscript. Nonetheless, since the incidence of FCS is not entirely known, the manuscript is of interest and explores ways of improving detection. He authors do a nice job of discussing factors which may help improve the FCS scoring system.

While the methods the authors utilize are acceptable for detection and clinical diagnosis of FCS, the study and its conclusions are limited by the lack of genetic confirmation of the diagnosis. This study would also benefit from a genetic risk score, which, in the presence of a homozygous or compound heterozygous gene variant, would significantly influence the clinical presentation. Furthermore, extensive genetic testing would be needed to also confirm that there were no triglyceride lowering gene variants in the study population. Both types of gene variants would affect their outcome. Although not included in this study, this should at least be mentioned in the discussion as a limitation.

The authors’ results are clearly in excess of published data, which they acknowledge. It is implied, but should be stated clearly that this study and the use of their model, may overestimate the incidence of this disease.

Author Response

Dear Reviewer, thank you for your valuable comments and suggestions on our manuscript. We hereby answer your recommendations as follows:

  1. A few grammatical changes would improve the flow and readability of their manuscript.

Answer: Thank you for your suggestion. We revised and restructured several sentences in our manuscript with a supervision of a native English speaker.

  1. While the methods the authors utilize are acceptable for detection and clinical diagnosis of FCS, the study and its conclusions are limited by the lack of genetic confirmation of the diagnosis. This study would also benefit from a genetic risk score, which, in the presence of a homozygous or compound heterozygous gene variant, would significantly influence the clinical presentation. Furthermore, extensive genetic testing would be needed to also confirm that there were no triglyceride lowering gene variants in the study population. Both types of gene variants would affect their outcome. Although not included in this study, this should at least be mentioned in the discussion as a limitation.

Answer: We absolutely agree with you, data of genetic analysis on variants either increasing or decreasing serum triglyceride levels would have improved the manuscript. For your kind information, we have started the genetic analysis of those 21 patients with the clinical diagnosis of FCS (depicted on figure 1); and screening for the mutations of the LPL, APOC2, APOA5, LMF1, GPIHBP1 and G3PDH1 genes are currently in progress. Completing the analysis of 9 individuals so far, we have found confirmed or potentially pathogenic mutations in 5 of them. We also agree with your comment on the genetic risk score; however, this would require extensive genetic testing, as you also mentioned.

Nevertheless, based upon your valuable comment and the suggestion of the other reviewer, lack of genetic analysis was also added to the limitations of our study in the discussion, as follows (lines 361-……):

“Also, we were unable to assess the data about family history and did not perform genetic testing to diagnose FCS. Verifying the existence of confirmed or potentially pathogenic mutations in LPL or other genes modulating lipoprotein lipase activity would have contributed to improve identification of potential FCS patients in the studied population. Genetic analysis of gene variants with triglyceride lowering effect would also have modified our results.”

  1. It is implied, but should be stated clearly that this study and the use of their model, may overestimate the incidence of this disease.

Answer: We thank you for your comment and completely agree with that. Our model of data mining and machine learning has a tendency to overestimate FCS frequency; however, it helps to reduce the number of such individuals that might require costly and time-consuming genetic testing.

Based upon your suggestion, we added the sentence in the discussion as follows (lines 369-….):

“While machine learning may overestimate the incidence of FCS, it also may help to reduce the number of those individuals that would require expensive and time consuming genetic analysis.”

Again, we are really thankful for your valuable comments on our manuscript. We think that your suggestion really helped to improve our manuscript.